

# Development of a distributed hybrid seismic-electrical data acquisition system based on NB-IoT technology

Wenhao Li [1], Qisheng Zhang [1*], Qimao Zhang [2], Feng Guo [1], Shuaiqing Qiao [1], Shiyang Liu [1], Yueyun Luo [1], Yuefeng Niu [1], Xing Heng [1].

[1]School of Geophysics and Information Technology, China University of Geosciences (Beijing), Beijing, China

[2]Institute of Electronics, Chinese Academy of Sciences, Beijing, China

**Correspondence:** Qisheng Zhang (zqs@cugb.edu.cn)

**Abstract.** The non-uniqueness of geophysical inversions, which is based on a single geophysical method, is a long-standing
problem in geophysical exploration. Therefore, multi-method geophysical prospecting has become a popular topic. In multi-method geophysical prospecting, the joint inversion of seismic and electric data has been extensively researched for decades. However, the methods used for hybrid seismic-electric data acquisition that form the base for multi-method geophysical prospecting techniques, have not yet been explored in detail. In this work, we developed a distributed, high-precision, and hybrid seismic-electrical data acquisition system using advanced Narrow Band-Internet of Things (NB-IoT) technology. The
system was equipped with hybrid data acquisition board, a high-performance embedded motherboard based on field-programmable gate array and advanced RISC machine, and host software. The data acquisition board used an ADS1278 24-bit analog-to-digital converter and FPGA-based digital filtering techniques to perform high-precision data acquisition. The equivalent input noise of the data acquisition board was only 0.5 μV with a sampling rate of 1000 samples-per-second and front-end gain of 40 dB. The multiple data acquisition stations of our system were synchronized using oven-controlled
crystal oscillators and global positioning system technologies. Consequently, the clock frequency error of the system was less than $10^{-9}$ Hz @ 1 Hz after calibration, and the synchronization accuracy of the data acquisition stations was ±200 ns. The use of sophisticated NB-IoT technologies allowed the long-distance wireless communication between control center and data acquisition stations. In validation experiments, it was found that our system was operationally stable and reliable, produced highly accurate data, and functionally flexible and convenient. Furthermore, using this system, it is also possible to
monitor the real-time quality of data acquisition processes. We believe that the results obtained in this study will drive the advancement of prospective integrated seismic-electrical technologies and promote the use of IoT technologies in geophysical instrumentation.

## 1 Introduction

Oil, gas, and minerals are extremely important to safeguard the development of the economic interests of a country and serve
as the material foundation of social development (Teng, 2007). After several decades of resource exploration and



exploitation, the energy resources on the surface of the Earth are diminishing, which is pushing China's resource exploration deeper into the Earth. It is a difficult task that will inevitably face many challenges in the future ((Qiao et al., 2011). Seismic and electrical exploration are the two most common geophysical methods currently used for prospecting. Seismic exploration is used to probe massive depths and is extremely precise and resolving (Van et al., 2001). Hence, it is widely

used in oil and gas industry, groundwater and geothermal resource exploration, and urban underground pipeline detection. However, its use in mining regions with poorly defined strata leads to dramatic lateral signal variations and low signal-to-noise ratios (SNR) (Mrmureanu et al., 2011). Electrical exploration has been rapidly developed and adopted in mineral resource prospecting, engineering surveys, and environmental surveys. However, these techniques are not sufficient to research the fine ground structures because of the limitations in vertical resolution of electrical exploration. Furthermore, the

non-uniqueness of single-method geophysical inversions has always posed a significant challenge to geophysicists. Multi-method joint exploration technologies have the potential to combine the strengths of each geophysical technique in a complementary manner, reducing the non-uniqueness of geophysical inversions, and thus improving the reliability of the conclusions drawn from geophysical exploration (Liu et al., 2012). Hence, these technologies have gained an intense interest in the realm of geophysical research and development. The integrated inversion of seismic and electrical data has been

recently investigated around the world, however, the hybrid seismic-electrical data acquisition techniques, which form the front-end of this approach, are yet to be probed in the research community.

The advancement in the realm of geophysics is closely related to the development of high-end geophysical instruments (Liu, 2017). The ceaseless development of advanced electronic technologies has led to tremendous progress in the instrumentation for seismic and electrical exploration. In the 1970s, Sercel (France) launched the SN338 digital seismograph and single-

station single-channel data acquisition station (Huang and Yu, 1994). In this particular backdrop, the most advanced land seismic data acquisition systems in the international market are the Sercel 508XT, Inova G3i HD, and Inova FireFly DR31 systems. As for electrical exploration instruments, the V8 system by Phoenix (Canada) and the GDP-3224 systems by Zonge (USA) are state-of-the-art electrical exploration instruments. These foreign seismic and electrical exploration instruments have achieved high levels of data accuracy, reliability, and smartization. In recent years, many companies in China have

been working in the development of seismic and electrical exploration instruments. In-depth studies have been performed on the topics related to channel capacity, data transmission methods (and rates) (Guo, 2012; Zhang, 2007), power consumption, networking modes, and data accuracy of seismographs (Wang, 2010; Zhang et al., 2013). In terms of their versatility, portability, stability, and commercialization, a number of important milestones have also been achieved in the development of electrical exploration instruments (Di et al., 2013; Lin et al., 2014; Chen et al., 2018; Cheng et al., 2019; Chen et al.,

2010). However, integrated seismic-electrical exploration instruments remain quite scarce, both in China and overseas. PASI (Italy) has launched its 16SG24-N combined system for seismic and electrical imaging, while the KMS (USA) has launched the KMS-850 data acquisition station. On the contrary, Chinese researchers have developed a multi-functional, high-power, multi-channel, rolling and fast measurement electrical method earthquake comprehensive measuring system (Zhang et al.,



2011). Nonetheless, the centralized data acquisition system cannot achieve high sampling densities and high accuracy, which are necessary for deep earth exploration.

Due to an increase in the number of data acquisition channels required by geophysical exploration and difficulties faced during the field exploration process, the need for high sampling densities over large areas in geophysical exploration can no

longer be met by centralized exploration instruments. As for distributed instrument systems, the productivity of wired systems is decreased by complex terrain. Whereas, wireless systems are very flexible in their usage as they do not need data transfer cables for their operation. The rapid development of Internet of Things (IoT) technology in recent years has led to the rise of short-distance communication protocols such as Zigbee and Bluetooth, which have short data transfer distances and complex network topologies. Long-distance communication protocols, such as 3G and 4G, have high data transfer rates

but place significant power consumption demands on their associated equipment (Li et al., 2011). Therefore, low-power wide area network technology was conceived to balance the weaknesses in these protocols. In particular, narrow band IoT (NB-IoT) is a cost- and power-efficient technology that supports a large number of connections, data transfer over the long distances, and deep coverage (Dong et al., 2017). These advantages have led to the development and application of NB-IoT in the realm of environmental monitoring, smart parking, and smart logistics (He et al., 2018). To enable the wireless control

and monitoring of distributed exploration instruments over the wide area, we used advanced NB-IoT technologies to develop a high-precision distributed and wireless hybrid seismic-electrical data acquisition system, which could be controlled and monitored in real-time. We believe that the proposed system will reduce the costs of field exploration, improve data quality, and drive the development of integrated seismic-electrical exploration technologies.

## 2 Data acquisition system design

The data acquisition system consisted of detectors, the hybrid seismic-electrical data acquisition board, field programmable gate array (FPGA), advanced RISC machines (ARM)-based embedded motherboard, and host software. A block diagram of the hardware components that make up the data acquisition station is shown in Figure 1. The detectors converted the non-electrical signals into electrical signals to enable their capture by the sampling circuit. The sampling board had eight sampling channels, four for seismic and four for electrical signals. The type and number of channels that were activated on

the sampling board could be adjusted according to the practical needs. As a results, the hybrid multichannel seismic-electrical data acquisition was enabled. A front-end conditioning circuit was used in the sampling board to remove interference from the analog signal. Afterwards, the analog-to-digital conversion (ADC) was performed for digital signal generation.

The motherboard is the core of the data acquisition station, because it performs a variety of tasks including data caching and

storage, state monitoring for data acquisition station, controlling the data acquisition system, and facilitating human-machine interactions. The motherboard of the presented system used a "FPGA+ARM" scheme. The ARM was a Texas Instruments AM4379 processor based on the ARM Cortex-A9 core, which is capable of handling frequencies of up to 1 GHz and was





operated with Linux. The FPGA was the 5CEBA7F23 chip from Intel's Cyclone V FPGA family, which is relatively cheap and has a rich-set of logical resources. The combination of these chips allows the data acquisition station to afford a wide range of functions and control over the system. An NB-IoT module was also included in the data acquisition system, which was used to enable the wireless communications between the system and the control center over long distances. The data

5 acquisition stations were monitored and controlled by the host software.

In addition, an ADUM1401 quad-channel digital isolator was added between the data acquisition board and motherboard to prevent interference and improve data transmission speed and efficiency. The ADUM1401 digital isolator uses iCoupler technology to perform digital isolation, and is free from the current transfer ratio instabilities and high-power consumption levels that plague opto-isolators. The system was powered by a 12 V power supply, where the voltage conversion was

10 performed by a high-efficiency DC-DC module (PTH08080WAH) to provide the power supply voltages required by each module.

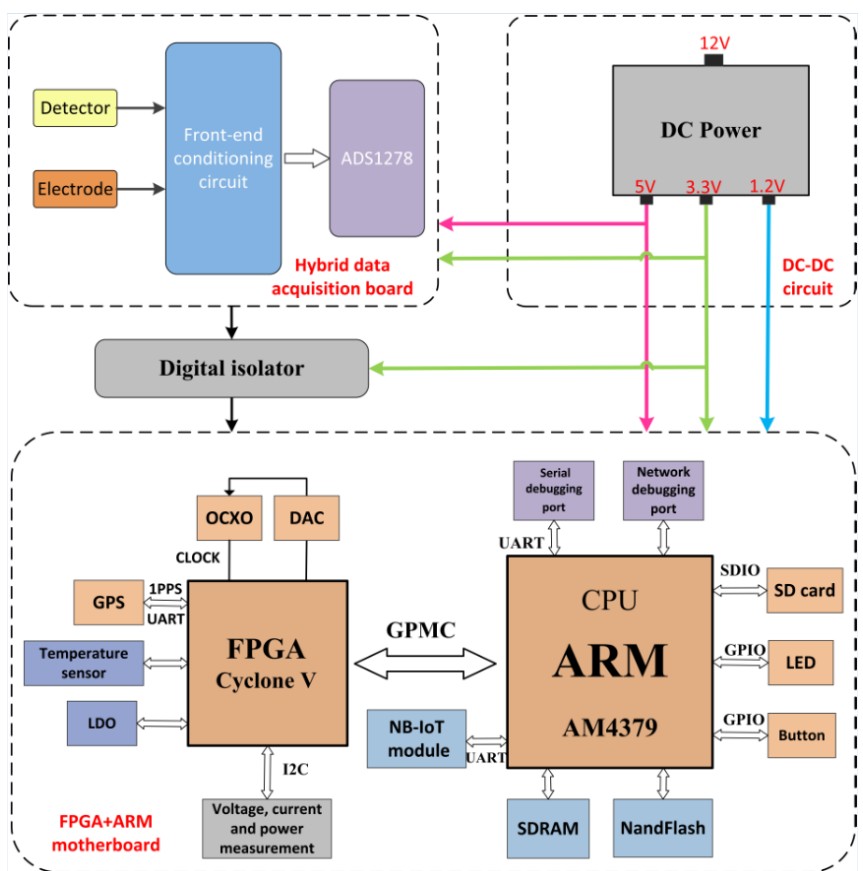

**Figure 1. Block diagram of the data acquisition station's hardware components**

**3 Key technologies in the design of data acquisition system**





### 3.1 Design and implementation of the High-precision sampling circuit

The sampling circuit is responsible for conditioning the analog signals acquired by the detectors and convert these signals into digital signals. A block diagram of the components of the sampling board is shown in Figure 2. The analog signals in each channel were first passed through a protection and resistivity matching circuit, which were then filtered and amplified.

Then, an ADC converted the analog signals into digital signals. The resulting digital signals were filtered again by the finite-infinite response (FIR) filter IP core in the FPGA.

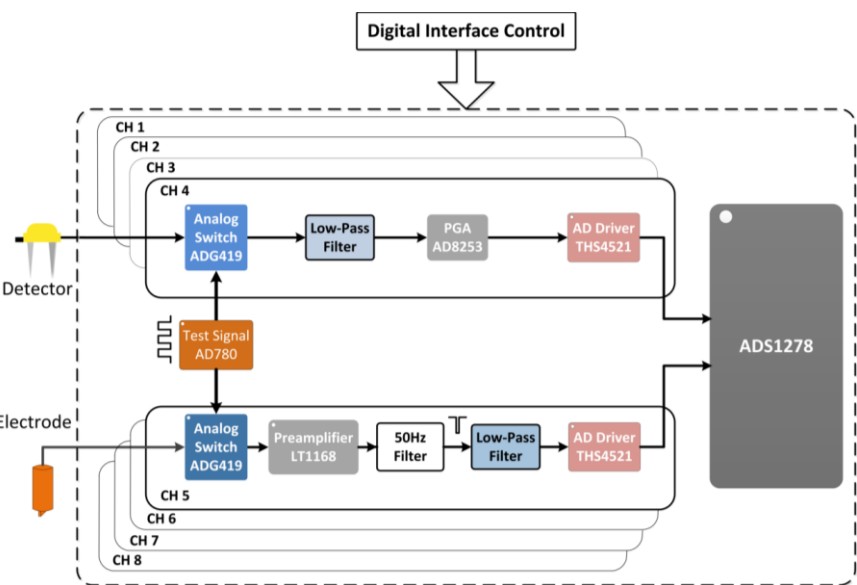

**Figure 2. Block diagram of the sampling board**

The front end of the seismic channel was connected to seismic detectors, and the analog signal given by the detectors were

passed through a passive low-pass filter to remove interference. Afterwards, the cleaned signal was amplified by an ADI AD8253 programmable gain instrumentation amplifier. The AD8253 had four levels of gain (0/20/40/60 dB), where it was possible to adjust the gain according to the amplitude of the acquired signals, and the sampling data were enabled with high levels of precision. Given that the ADS1278 ADC input signals are required to balance differential signals, a fully differential THS4521 amplifier was inserted between the programmable amplifier and the ADC, as shown in Figure 3. The

THS4521 amplifier converted the single-ended output of the programmable amplifier into a differential signal and was powered by a +5 V line. The +2.5 V common mode voltage was supplied by a high-precision REF5025 voltage reference. Then, the analog signals were converted into digital signals by a 24-bit high-precision ADS1278 ADC. The digital signals were then filtered by the FIR filter IP core in the FPGA to further improve the SNR of these signals.



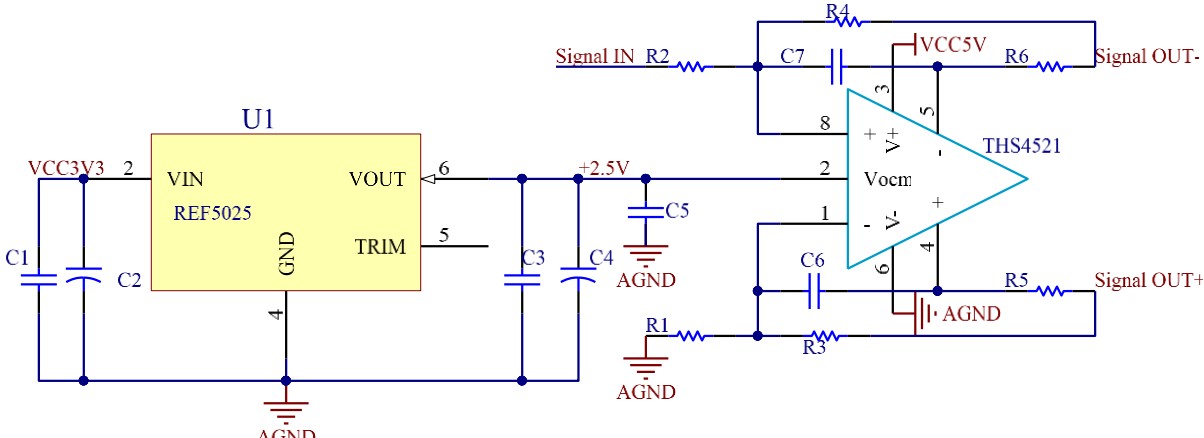

**Figure 3. Circuit diagram of the AD driver.**

The signal processing methods used in the electrical channel were similar to those of the seismic channel. The front end of the electrical channel was connected to electrodes, where the low-pass filtering and amplification of the analog signals were

5 performed by the conditioning circuit. For signal amplification, a low-power LT1168 precision instrumentation amplifier was used. Furthermore, the amplifier gain was adjusted using a single resistor. Given that the interference of the frequency of the mains (50 Hz) has a significant impact on electrical exploration, the interference due to the 50 Hz component was reduced using a combination of analog filters and FPGA digital filtering. The OPA4227 operational amplifiers (opamps) were used to construct a mains frequency filtering circuit, as shown in Figure 4. In this circuit, three opamps were used to

10 construct a bandpass filter, while the fourth opamp was used to subtract the output of the bandpass filter from the all-pass circuit, which eliminated the 50 Hz component.

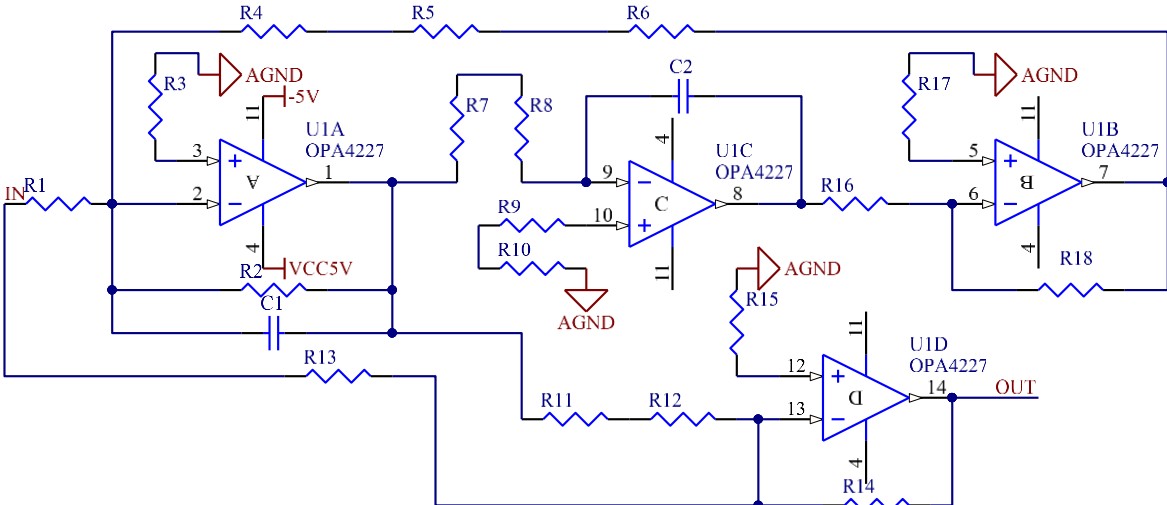

**Figure 4. Circuit diagram of the mains (50 Hz) interference filter.**



## 3.2 High-performance embedded main control system design

The main control system uses an AM4379 processor operating in Linux 3.14.43 and a Cyclone V programmable logic device, which provide a variety of functions, as shown in Figure 5.

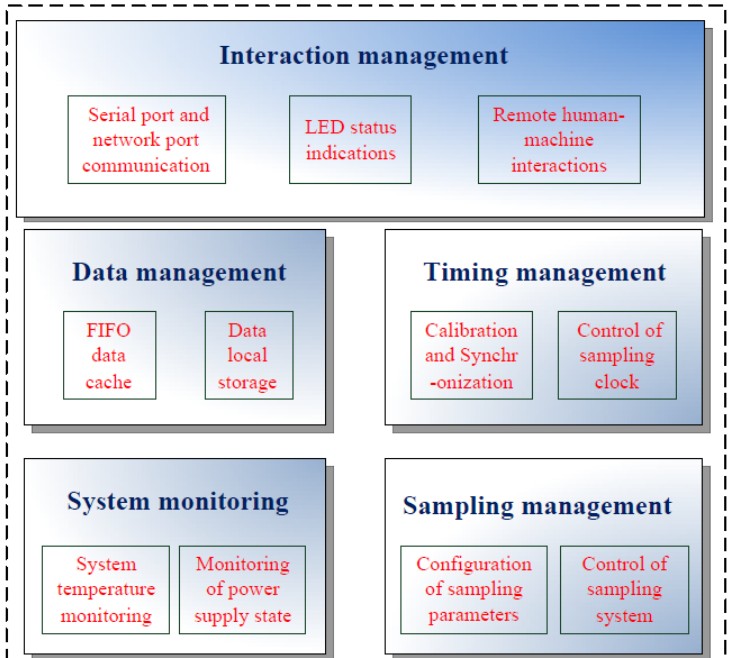

**Figure 5. Primary functions of the main control system.**

Herein, the FPGA communicated with the ARM processor through a general-purpose memory controller (GPMC) bus. The FPGA was programmed using the Quartus II development environment. As the FPGA is highly programmable and has powerful timing functions, it played an important role in the main control system of the data acquisition station. The prime FPGA functions include: (1) The supply of multiple sampling clocks to the data acquisition system to enable sampling at

different rates, (2) controlling and recording data from the LTC2945 voltage/current sensor and DS18B20 temperature sensor using Verilog HDL, (3) controlling the First Input First Output(FIFO)caching of seismic-electrical data, the number of open channels in the sampling board, the gain on each channel, and the sampling rate, and (4) receiving and parsing information from the GPS module to calibrate the oven-controlled crystal oscillator (OCXO) and synchronize the data acquisition stations using the pulse per second (PPS) signal of the GPS module.

Because of an abundance of peripheral interfaces, the AM4379 is considered as a very powerful processor. In Figure 1, it has already been depicted that the ARM processor communicates with the SD card used for local data storage through its secure digital input output (SDIO) interface and controls the light emitting diodes (LEDs) and buttons through the general-purpose input/output (GPIO) interface. The LED lights indicated the operational status of the system, while the buttons allowed the manual control of the data acquisition station. A high-performance LAN922Li-ABJZ ethernet controller was used to provide

a 10/100M wired network interface, which could be used to debug the station or data collection if it is not possible to remove


the SD card. The Quectel BC95 NB-IoT module, which has very low power consumption, was used in this system. The NB-IoT module facilitated wireless communications between the data acquisition station and control center, and was connected to the ARM via a universal asynchronous receiver-transmitter (UART).

The embedded software of the main control system was designed using a modular top-down approach. The embedded

software was divided into four submodules and one main task according to the functions of the data acquisition station, where each functional module was implemented using a single thread to maximize the responsiveness of the software. The main thread was responsible for self-diagnostics, equipment initialization, and the seismic and electrical data acquisition. The status monitoring thread was responsible for monitoring all information related to the operational status, temperatures, voltages, currents, SD card storage space, and GPS synchronization of the data acquisition station. To report the status of the

data acquisition station and receive commands from the control center, the wireless communications thread was responsible for remote interactions with the control center via the NB-IoT module. The LED indicator thread indicated the current operational status of the data acquisition station by controlling the blinking rate of certain LEDs and the number of blinking LEDs. The button management thread identified and analyzed the user's button operations, and allowed the station to respond in accordance to the button operations. During the design of this software, hardware control was performed by

adjusting the interface functions of the system's standard libraries, where the relevant functions were implemented.

### 3.3 High-precision clock and synchronization performance

Seismic exploration requires tight synchronization between the data acquisition stations, which makes a highly stable and precise clock a necessity for station synchronization. The OCXO used in our design had a nominal output frequency of 12.288 MHz. The short-term stability of the OCXO was $\leq \pm 2 \times 10^{-11}/S$, and its aging rate was $\pm 1 \times 10^{-7}$. After one year of

aging, the estimated daily cumulative deviation of the instrument would become 13 ms, thus making it necessary to calibrate the OCXO. The principles of calibration of the OCXO are illustrated in Figure 6. The frequency of the OCXO was measured by the FPGA via a PPS signal generated by the GPS receiver as a time gate. Then, the deviation of the OCXO's frequency and the corresponding calibration offset were calculated. The required voltage adjustment was calculated according to the calibration offset, and the corresponding digital signal was converted into an analog signal by a digital to analog converter

(DAC). The analog signal was then passed to the voltage control terminal of the OXCO to calibrate the OCXO.

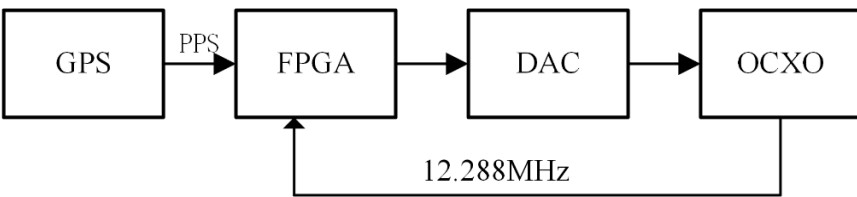

**Figure 6. A flow chart representation of OCXO calibration.**



The OCXO calibration was validated by measuring its output frequency via a 10 MHz clock of a chip-scale atomic clock
(which has a frequency accuracy of $10^{-10}$) as the reference clock source. The frequency of the OCXO prior to calibration was
12.28799928 MHz, which corresponds to an accuracy of:

$$\frac{12.288 - 12.28799928}{12.288} \approx 5.86 \times 10^{-8} \tag{1}$$

After calibration, the OCXO frequency was 12.28799995 MHz, which corresponds to an accuracy of:

$$\frac{12.288 - 12.28799995}{12.288} \approx 4.07 \times 10^{-9} \tag{2}$$

After the OCXO calibration, its frequency accuracy was in the order of $10^{-9}$. The daily cumulative deviation of the OCXO
was now less than 1 ms, which was sufficiently accurate for long-term data acquisition.

Given that the synchronization in the data acquisition stations allows their signals to be synchronized with GPS PPS signals,

the PPS signals were compared to a third-party GPS PPS signal to examine the time accuracy of synchronized sampling by
the data acquisition stations. The results of aforementioned test are shown in Figure 7. Each division (dotted lines) of the
time axis (horizontal axis) corresponds to 100 ns. Channel 1 (blue) is the PPS signal of the reference TGP2-35 GPS system,
while Channels 2 (green) and 3 (purple) are the PPS signals of two data acquisition stations. The synchronization accuracy of
the data acquisition stations is within 200 ns when GPS lock is achieved. This level of synchronization was sufficient for

synchronized data sampling.

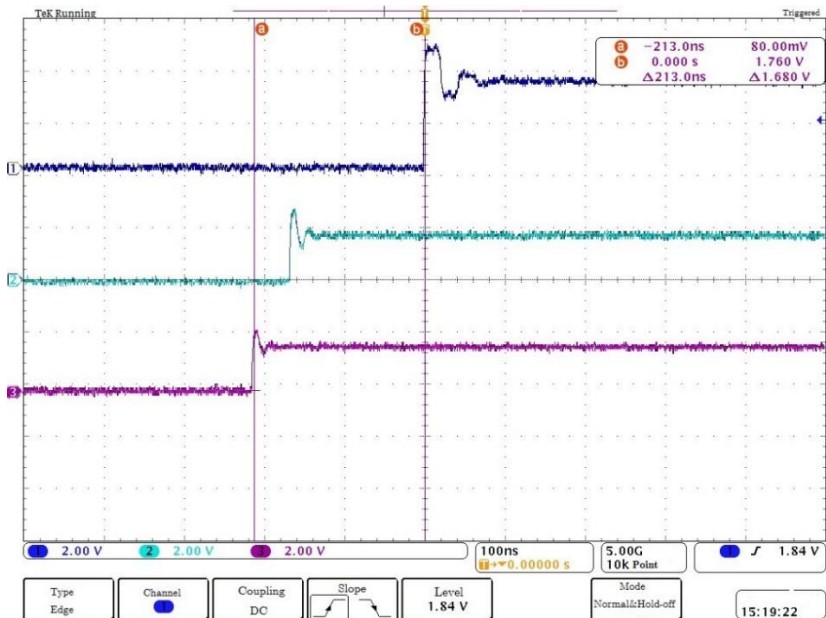

**Figure 7. Synchronization accuracy examination of the data acquisition stations.**

## 3.4 NB-IoT communications





The NB-IoT allowed the data acquisition stations to report their operational status to the control center, and the control center to issue commands to the stations. A flow chart of the communication between data acquisition stations and the control center is shown in Figure 8. Given that single-carrier transmission reduces the power consumption, NB-IoT uplink transmissions were performed using the single-carrier frequency-division multiple access scheme. The data transfer speed of

the NB-IoT uplink was approximately 200 kb/s. The downlink transmissions were modulated by the quadrature phase-shift keying scheme, which was compatible with long-term evolution downlinks. The data transfer speed of the downlink was the same as that of the uplink.

The NB-IoT communications system consisted of a device layer, network layer, IoT platform layer, applications layer, and management layer. The device layer was the data acquisition station which was equipped with the NB-IoT module. It was

used to pass the various forms of information that were being monitored by the NB-IoT base station (the network layer) via an air interface. The NB-IoT base station handled the processing of the air interface, which was connected to the core network through an S1 interface. Because of its simple architecture which better suits low-speed and wide coverage networks, this core network was different from the current evolved packet core network. The interactions with the terminal non-access stratum were handled by the core network. The IoT platform layer was based on the China Telecom IoT Open

Platform, which collected the IoT data from various access networks and sent each type of data to their corresponding application services for processing. The power supply monitoring, GPS querying, data acquisition control, storage status, map display, and temperature display application program interfaces were provided by the applications layer, which enabled the monitoring, management, and control of each data acquisition station via a control center using computer or mobile apps.

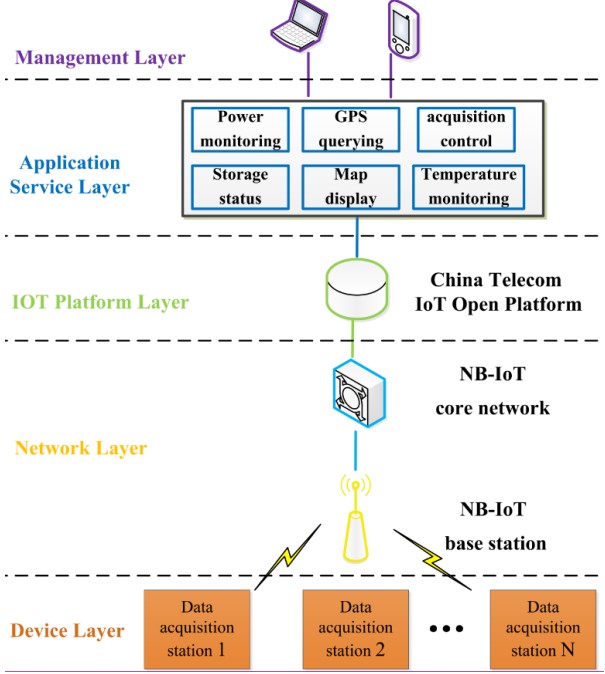

**Figure 8. A flow chart representation of the NB-IoT communications.**



## 4 Host software design of the control center

Based on the transmission control protocol (TCP) and user datagram protocol (UDP), the host software was developed using the object-oriented C# programming language, where the software had three parts. The first part monitored the status of each data acquisition station, including power supplies (voltage and current), data acquisition in progress/terminated, storage, and

synchronization. The process allowed the data acquisition stations to be inspected and maintained during operational processes. The second part controlled the data acquisition stations and configured their sampling parameters, such as the gain and sampling rate of each channel. It allowed the data acquisition stations to be controlled and configured remotely and to suit the data acquisition process requirements in different environments. In the third part, the data acquisition stations locations were displayed using the GPS coordinates reported by the stations. To make it easy to patrol each station, the data

acquisition stations' locations were displayed on a map. A part of the host software interface is shown in Figure 9.

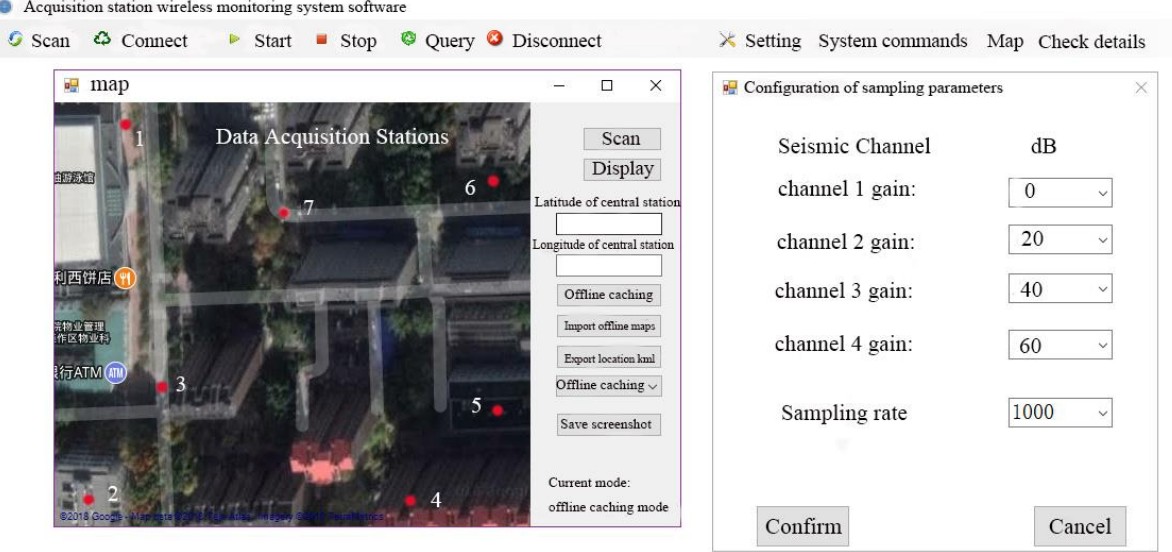

**Figure 9. A part of the host software interface**

## 5 System tests

### 5.1 Input noise and dynamic range tests

The equivalent input noise (EIN) consisted of many types of noises, where the metric was used to determine an instrument's ability to resolve weak signals. In these tests, a 1 kΩ resistor was connected to the signal input terminal, and the sampling rate was set to 1000 SPS. When the front-end gains of the seismic channels were set to 0 dB, 20 dB, 40 dB, and 60 dB and the gains of the electrical channels were set to 0 dB, 10 s of data sampling was performed at each station. Furthermore, acquired data were analyzed with MATLAB. Figure10 shows distribution of the EIN of a data acquisition station with a

sampling rate of 1000 SPS and gain of 60 dB.



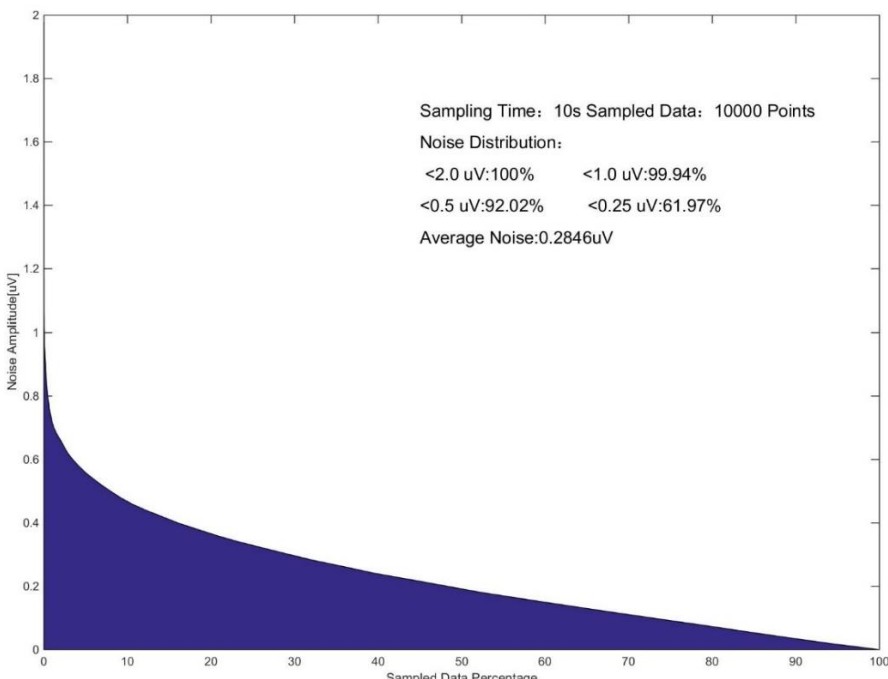

**Figure 10. Distribution of the EIN of a data acquisition station.**

The EIN test results with different channels and gains are shown in Table 1. When the front-end gain of the system was 40 dB, the EIN of the sampling board was 0.50 μV, which meets the requirements of field exploration works.

5          **Table 1. Results of EIN tests.**

| Channel type | Amplifier gain (dB) | Equivalent input noise (μV) |
|---|---|---|
| Seismic channel | 0 | 5.92 |
| | 20 | 0.82 |
| | 40 | 0.50 |
| | 60 | 0.28 |
| Electrical channels | 0 | 7.12 |

The ratio between the strongest and weakest signals that can be acquired by an instrument without distortion is the dynamic range of the instrument (with the weakest signal usually corresponding to the noise level of the instrument), and this range is usually expressed in decibels (dB). The dynamic range test was performed with a sampling rate of 1000 SPS and gain of 0 dB. A signal generator was used to produce a 1 Vpp 31.25 Hz sinusoidal signal, and the amplitude of this signal was

10    continuously increased until the output signal reached its maximum amplitude. By calculating the ratio between the maximum signal level and noise level of our system, it was found that the dynamic range of the sampling board was 107 dB.

**5.2 Channel crosstalk tests**




Channel crosstalk is inevitable because of the presence of many channels in sampling boards. During the channel crosstalk tests, Channels 3 and 4 of the data acquisition station were shorted while a 1 kHz sine wave with a peak voltage of 4 V was fed into the Channels 5 and 6 .The gain of all channels was set to 0 dB and data acquisition was performed for a period. Fast Fourier transforms (FFT) were then performed on the collected data to compare the signal amplitude ratios of the channels.

The results of the channel crosstalk test are shown in Figure 11, where the red and blue plots are the FFTs of Channels 4 and 6, respectively. This figure shows that the power of the input signal at 1 kHz is 8.073 dB (relative value), and the 1 kHz peak is buried under the noise in the red plot (below -128 dB).

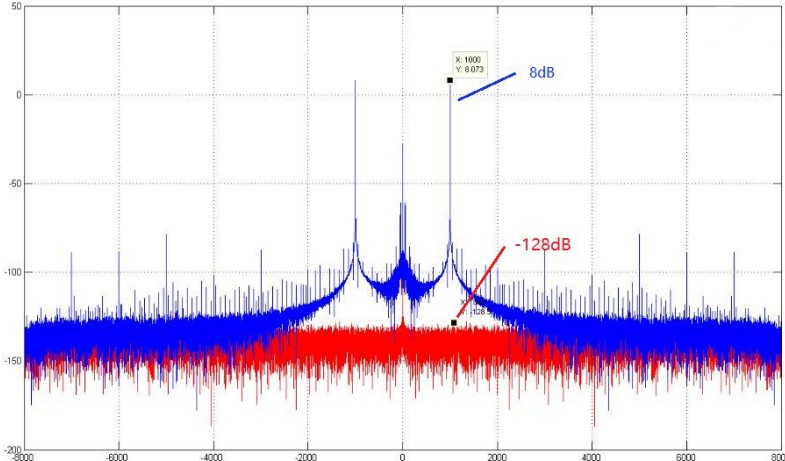

**Figure 11. Comparison between the signal spectra of channels 4 and 6 (32768-point FFT).**

The 1 kHz crosstalk signal in Channel 4 began to appear when FFT calculations with a larger number of points (1288192 points) were performed on each channel's data, as shown in Figure 12. However, the crosstalk signal power (-119.2 dB) was much smaller than the signal's power (23.98 dB). This indicates that the level of crosstalk between the channels satisfies design requirements.

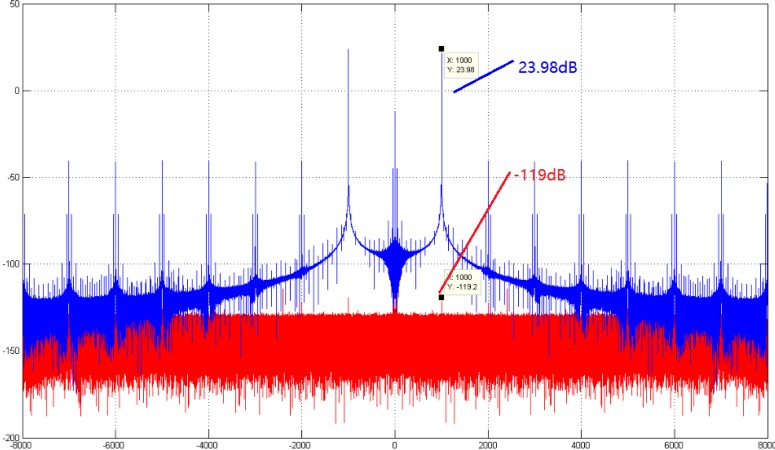

**Figure 12. Comparison between the signal spectra of channels 4 and 6 (1288192-point FFT).**





### 5.3 Test of communications between the data acquisition stations and the control center

After the design and testing of the system's software and hardware components, the data acquisition stations' ability to communicate with the control center was examined in Beijing's suburbs. The stations were laid out in a distributed manner, where each station was monitored and managed by a control center. Figure 13 shows the status data that was uploaded to the control center via a NB-IoT system during the sampling processes of the data acquisition stations. This data allowed the management personnel to maintain and manage each data acquisition station in a targeted manner, which helped to improve operational efficiency and safeguard the quality of the acquired data. Furthermore, the gain, sampling rate, and operational status of each data acquisition station were adjusted by the control center. It was found that the command delivery rates of our system were adequate for practical applications.

| Station number | Operational status | Number of samples | Dropped packets | Synchronization status | Device temperature | Remaining storage space | Battery voltage | Battery current | GPS satellite number |
|---|---|---|---|---|---|---|---|---|---|
| 1901 | Acquiring | 20000 | 0 | Synchronized | 35.4℃ | 28673.19 MB | 10.70 V(10%) | 0.39 A | 5/10 |
| 1902 | Acquiring | 20000 | 0 | Synchronized | 36.9℃ | 30319.66 MB | 10.87 V(25%) | 0.41 A | 6/11 |
| 1903 | Acquiring | 20000 | 0 | Synchronized | 35.6℃ | 29045.00 MB | 11.32 V(55%) | 0.40 A | 7/9 |
| 1904 | Acquiring | 20000 | 0 | Synchronized | 36.7℃ | 29051.34 MB | 12.27 V(100%) | 0.38 A | 6/9 |
| 1905 | Acquiring | 20000 | 0 | Synchronized | 34.1℃ | 29072.03 MB | 10.35 V(10%) | 0.43 A | 7/8 |
| 1906 | Acquiring | 20000 | 0 | Synchronized | 35.9℃ | 30020.59 MB | 12.25 V(100%) | 0.37 A | 9/9 |
| 1907 | Acquiring | 20000 | 0 | Synchronized | 36.2℃ | 30174.28 MB | 11.25 V(50%) | 0.38 A | 8/10 |
| 1908 | Acquiring | 20000 | 0 | Synchronized | 34.9℃ | 29051.11 MB | 12.22 V(100%) | 0.39 A | 10/14 |
| 1909 | Stopped | 0 | 0 | Synchronized | 36.4℃ | 30173.19 MB | 11.85 V(85%) | 0.35 A | 7/13 |
| 1910 | Stopped | 0 | 0 | Synchronized | 35.2℃ | 30051.32 MB | 10.87 V(25%) | 0.41 A | 6/8 |

**Figure 13. Screenshot of the status data reported by each data acquisition station.**

### 6. Conclusion

Herein, the study develops a distributed hybrid seismic-electrical data acquisition system. The paper mainly explores the following technical aspects:

(1) NB-IoT technology was used to achieve long-distance wireless communication between the control center and data acquisition stations, which facilitated the data acquisition stations' monitoring and control. It was conducive to ensuring a high level of data quality and operational efficiency.

(2) High-precision hybrid seismic and electrical data sampling was achieved using an eight-channel 24-bit ADS1278 ADC and FIR digital filtering. With a sampling rate of 1000 SPS and gain of 40 dB, the EIN of the sampling system was only 0.5 μV.

(3) The OCXOs and GPS technology were used to synchronize multiple data acquisition stations. The clock error of the calibrated OCXO was less than $10^{-9}$ Hz @ 1 Hz, and the synchronization accuracy of the data acquisition stations was ±200 ns.

(4) We believe that the propounded system can be used to conduct the integrated geophysical exploration of underground urban spaces using seismic and electrical data. Furthermore, the development of this system will promote the application of IoT technologies in geophysical instrumentation.



*Author contributions.* The author worked as the hardware design and post-debugging throughout the development process, as well as the drafting of the manuscript.

*Acknowledgements*. This work was supported by the Natural Science Foundation of China (No.41574131), the National Key
Research and Development Program of China (No.2017YFF0105704), the National "863" Program of China (No. 2012AA06110203), and the Fundamental Research Funds for the Central Universities of China.

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
