# Peer review of "Development of a distributed hybrid seismic-electrical data acquisition system based on NB-IoT technology"

_Geoscientific Instrumentation, Methods and Data Systems, 2018_

## Referee Comment (RC1) · Anonymous Referee #1 · 8 May 2019

The idea of this manuscript is original and unique, and it describes the technical realization of a new distributed data acquisition system for geophysical exploration. This is the first time I have seen the application of NB-IoT technology in geophysical instruments.   I firmly believe that the results obtained in this study will drive the advancement of prospective integrated seismic-electrical technologies and promote the use of IoT technologies in geophysical instruments. The findings of this study are very suitable for the GI journal. The quality of the figures in this manuscript is good. I have attached below several comments arising from the manuscript, which might be helpful to further improve the quality of the publication of the paper.  (1)The format of some of the references should be properly adjusted to ensure that the article format

is more standardized. (2)The Figure 3 "Circuit diagram of the AD driver" may not be needed because the circuit is not complicated, and it has been clearly expressed in words. (3)A final checking for any missed spelling errors may be necessary. Since the manuscript is left hanging for quite some time this process should come to an end and the paper finally being published. I am looking forward to the final publication.

Please also note the supplement to this comment:
https://www.geosci-instrum-method-data-syst-discuss.net/gi-2018-51/gi-2018-51-RC1-supplement.pdf

---

## Referee Comment (RC2) · Anonymous Referee #2 · 20 May 2019

The manuscript describes the technical realization of a distributed data acquisition system that combines both seismic and electrical methods of geophysical exploration. What's more, NB-IoT technology is applied to the developed instrument, and the idea is amazing. Examples of new technologies being applied to geophysical instruments are rare, so the study should be encouraged and supported. I think the findings of this study are very suitable for the GI journal for Geoscientific Instrumentation. I am looking forward to the final publication of the paper.

I have read the manuscript carefully and attached a few comments. 1. In the "5.1 Input noise and dynamic range tests", it's unnecessary to explain the definition of dynamic

range because it is basic knowledge. 2. Figure 11 and Figure 12 are similar, and they are both the test results of channel crosstalk. Figure 11 has shown that the crosstalk signal power is much smaller than the signal's power, and it's better to verify the test result using FFT calculations with a larger number of points, but putting Figure 12 in the paper is redundant and repetitive. 3. The manuscript is written in an adequate style, and a final checking for any missed spelling or grammar errors may be necessary.

---

## Author Comment (AC1) · 1 Jul 2019

Thank you very much for your affirmation of my work and suggestions for this manuscript.  (1)I have adjusted some of the references to ensure the article format is more standardized.(2)As you said, Figure 3 is too simple and is superfluous in the paper.So,I have deleted it.(3)In order to avoid spelling mistakes, I read the manuscript carefully again.Thank you again for your precious comments.

---

## Author Comment (AC2) · 1 Jul 2019

Thank you very much for your comments on my article and for the recognition and encouragement of my work.(1)As you said,the definition of dynamic range is basic,and it's superfluous in the article.So I have deleted the definitoin.(2)I have removed Figure 12 to avoid repeatability because the Figure 11 has shown the test result of channel crosstalk.(3)I have read the manuscript again carefully to avoid missed spelling mistakes.Thanks again for your suggestions.